# SymCL: Riemannian Contrastive Learning on the Symmetric Positive Definite Manifold for Visual Classification

## Abstract

Symmetric Positive Definite (SPD) matric has been proven to be an effective feature descriptor in the realm of artificial intelligence, as it can encode spatiotemporal statistical information of data on a curved Riemannian manifold, *i.e.*, SPD manifold. Although existing Riemannian neural networks have demonstrated superiority in many scientific fields, the inherent reliance on labels within supervised learning renders them susceptible to label errors. Besides, it is insufficient to depend solely on labels to learn effective feature distributions in some complicated data scenarios. Drawing inspiration from the considerable achievements of contrastive learning (CL) across diverse tasks, we extend the conventional CL paradigm to the context of SPD manifolds, which we denote SymCL, paving the way for a novel approach in SPD matrix-based visual classification. Furthermore, we inject a Riemannian triplet loss-based Riemannian metric learning (RML) into the designed SPD manifold CL framework for the sake of improving the discrimination of the learned geometric representations. Extensive experimental results on four datasets verify the effectiveness of the proposed algorithm.

## 1 Introduction

In the field of pattern recognition and computer vision, deep learning technique has achieved significant progress in extracting powerful feature representations. These techniques are particularly adept at handling complex data structures, which is crucial for extending their application to specialized domains such as geometric data analysis. Especially in processing Symmetric Positive Definite (SPD) manifold data, deep learning models have made considerable progress.

However, the inherent difficulty in processing and classifying these matrices, which are essentially SPD, is that they cannot be regarded as Euclidean data points. This is because the topology formed by a group of SPD matrices of the same dimensionality is not a vector space, but a curved Riemannian manifold, *i.e.*, SPD manifold Arsigny et al. (2007). Therefore, it is inappropriate to directly compute SPD data points in Euclidean space. To address this limitation, Pennec et al. (2006) and Arsigny et al. (2007) employ Riemannian metrics to characterize the Riemannian geometric of Symmetric Positive Definite matrices (SPD), including Log-Euclidean Metrics (LEM) Arsigny et al. (2007) and Affine-Invariant Riemannian Metrics (AIRM) Pennec et al. (2006). These Riemannian metrics allow for the extension of Euclidean tools to the SPD manifolds. Specifically, this involves mapping the SPD manifold-valued data points into a flat space via tangent approximation Tosato et al. (2010); Sanin et al. (2013) and Tuzel et al. (2008), then embedding it in a Reproducing Kernel Hilbert Space (RKHS) that incorporates Riemannian kernel functions Harandi & Salzmann (2015); Wang et al. (2012; 2022a; 2015) and Harandi et al. (2012). Unfortunately, both of these methods primarily operate in Euclidean space for representation learning and classification, which inevitably distorts the geometric structure of the original data manifold. To counter this challenge, recent SPD matrix discriminant analysis methods have recently been proposed for geometry-aware feature transformation. Gao et al. (2019); Huang et al. (2015); Zhou et al. (2017); Nguyen & Yang (2023) and Chen et al. (2023). The core of these methods is to generate a low-dimensional feature manifold with high discriminability by simultaneously learning an embedding mapping and a similarity metric on the original SPD manifold. As a result, the resulting feature space can faithfully reflect the geometric structure of the input SPD data points.

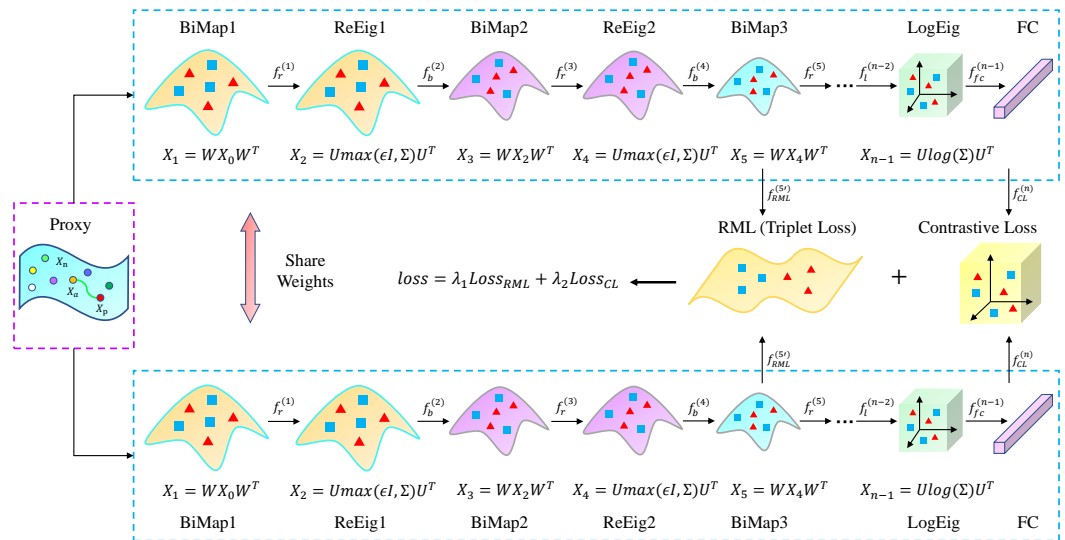

Figure 1: An overview of our proposed contrastive learning framework. Both the upper and lower branches of the framework are the same three-layer SPD network, consisting of a stack of BiMap and ReEig layers, followed by the LogEig layer for the embedding of manifold features into an Euclidean space. The upper branch processes anchor samples and positive samples, while the lower branch processes anchor samples and negative samples. Following feature extraction via the designed SPD contrastive network, we simultaneously calculate the contrastive loss and an additional RML loss, which is integrated at the end of the third BiMap layer, jointly supervising network training.

Although the aforementioned classification methods based on SPD manifold learning have achieved fruitful results, the inherent supervised learning paradigm makes the feature learning process over-reliance on semantic labels. The ablation study in the following section demonstrates that erroneous labels involved in the training process can have a negative impact on the final classification performance. In such a circumstance, how to empower the learning system to capture the intrinsic data structures and patterns without relying on the semantic labels is of great significance to the Riemannian manifold-based visual classification.

Contrastive learning (CL), a powerful self-supervised learning approach, has made significant progress in computer vision and natural language processing. Compared to traditional supervised learning paradigm, CL can analyze the intrinsic data distributions by measuring the similarity of a pair of samples in the metric subspace without requiring additional annotation information.He et al. (2020); Chen et al. (2020). Moreover, the study in He et al. (2020) demonstrates that features generated by CL are generally more roust and effective, and can be easily transferred to other tasks. Therefore, we generalize the potential and efficiency of CL to the scenario of SPD manifolds.

In this paper, the proposed SPD manifold-based CL framework mainly contains three parts, which are the Riemannian data augmentation(RDA), SPD encoder, and Riemannian triplet loss, respectively. It's well known that data augmentation (DA) is a crucial part of contrastive learning. However, due to the unique geometric structure of SPD matrices, traditional DA methods for Euclidean space (such as rotation, color change, crop) are no longer applicable. Therefore, using the properties of SPD matrix and the corresponding Riemannian operator, we design several DA methods for the proposed SymCL. We use SPDNet Huang & Van Gool (2017) as the encoder, and constructing positive and negative sample pairs within each batch to train SymCL. Specifically, in each batch, we randomly select one sample as the anchor point and generate a positive sample through DA, with the remaining samples serving as negatives. Then these samples are feed into the encoder for feature transformation mapping, utilizing the commonly used contrastive loss, infoNCE He et al. (2020), as the supervisor of the proposed model. Furthermore, we design a Riemannian triplet loss (based on Wang et al. (2022b)) for the sake of encoding and analyzing the geometric distribution of the

input data, explicitly. Besides, it is trained in conjunction with the infoNCE loss, forming a dynamic complementary relationship in feature representation learning.

In summary, the **main contributions** of this paper contain the following four aspects:

- **Generalizing the Euclidean CL paradigm to the SPD manifolds.** We explore the feasibility of self-supervised CL mechanism on the Riemannian manifold, opening up a new paradigm for SPD matrix learning.

- **A novel Riemannian triplet loss.** We introduce a Riemannian triplet loss to the proposed SymCL. This module can explicitly inject the encoding and learning of data distribution into the network training process, being potentially to induce more effective features.

- **Designing four Riemannian data augmentation methods.** We explore and demonstrate the feasibility of proxy in the context of Riemannian manifolds, and propose four Riemannian data augumentation schemes, facilitating the research of Riemannian CL.

- **Empirical validation on four benchmarking datasets.** Extensive experimental results obtained on four benchmarking datasets demonstrate the effectiveness of our method.

## 2 RELATED WORK

### 2.1 SPD MATRIX LEARNING

Traditional geometry-aware Riemannian learning techniques have made significant progress in processing and classifying visual data based on SPD matrices. However, their inherent shallow linear embedding mechanism proves inadequate for extracting fine-grained geometric features, particularly in complicated visual scenarios. Taking inspiration from the benefits of convolutional neural networks (ConvNet), the work studied in Huang & Van Gool (2017) creates an end-to-end Riemannian neural network that specializes in nonlinear learning of SPD matrices. Diverging from ConvNets He et al. (2016); Simonyan & Zisserman (2014) in feature learning and parameter optimization, this network takes structured SPD matrices as inputs, requiring each layer to maintain the Riemannian geometry of the data manifold. Therefore, the optimization of parameters also needs to be performed on the Riemannian manifold to ensure that the output is an SPD matrix.

### 2.2 CONTRASTIVE LEARNING

Contrastive learning can be tracked back to Hadsell et al. (2006). In this method, representations are learned by contrasting positive pairs against negative pairs. Inspired by this ideology, the work studied in Dosovitskiy et al. (2014) suggests treating each instance as a class, represented by a feature vector in a parametric form. Subsequently, another study work Wu et al. (2018) suggests to utilize a memory bank to store instance class representation vectors, this approach has adopted and extended in several recent papers Zhuang et al. (2019); Tian et al. (2020); He et al. (2020); Misra & Maaten (2020). Other works explore the utilization of in-batch samples for negative sampling instead of a memory bank Doersch & Zisserman (2017); Ye et al. (2019); Ji et al. (2019).

A common way to define a loss function is to measure the difference between a model's prediction and a fixed target. Contrastive loss, as described in Hadsell et al. (2006), quantifies the similarities between pairs of samples in a representation space. Unlike methods where inputs are matched against fixed targets, contrastive loss allows the target to dynamically vary during training and can be defined based on the learned representations Hadsell et al. (2006).

## 3 METHOD

Numerous experimental studies have shown that compared to Riemannian shallow learning techniques, Riemannian deep learning is capable of extracting more effective geometric features for improved visual classification. To the best of our knowledge, a majority of existing algorithms for learning SPD matrices are based on the supervised learning paradigm, which may impact their generalization ability. For instance, in complex data scenarios, there usually exist a wide range of intra-class diversity and inter-class ambiguity, rendering the traditional discrimination analysis approach that explicitly relies on label information fail to capture more realistic data distribution. Our

experimental findings indicate that the incorrect semantic labels can have a negative impact on the classification results. The issue of how to liberate models from reliance on the label semantics is a critical challenge that needs to be solved urgently in the domain of Riemannian manifold-based visual classification. In this article, we designed a self-supervised CL paradigm on the SPD manifold to open a new direction for the learning of SPD matrices.

## 3.1 PRELIMINARY

A family of $d$-by-$d$ SPD matrices is a commutative Lie group, with a manifold structure denoted as $S_{++}^d$. More formally:

$$S_{++}^d := \{X \in \mathbb{R}^{d \times d} : X = X^T, \nu^T X \nu > 0, \forall \nu \in \mathbb{R}^d \setminus \{0_d\}\}. \tag{1}$$

Therefore, the concepts of differential geometry, such as geodesic, can be utilized to address $S_{++}^d$. Moreover, any bi-invariant metric $\langle , \rangle$ on the Lie group of SPD matrices corresponds to an Euclidean metric in the SPD matrix logarithmic domain, i.e., the tangent space at identity matrix $T_I S_{++}^d$ (Please refer to the appendix A.1 for details.), it is also called the Log-Euclidean Metric (LEM), which can be formulated as:

$$D_{LEM}(X_i, X_j) = \|\log(X_j) - \log(X_i)\|_F. \tag{2}$$

The LEM works directly in the logarithmic domain of SPD matrices, offering higher computational efficiency. Therefore, we choose it as the distance metric in this paper.

## 3.2 PROXY - DATA AUGMENTATION

One of the core parts in CL is the DA mechanism. SimCLR Chen et al. (2020) has extensively researched DA strategies within traditional CL. However, the unique geometric structure of SPD matrices determines that the aforementioned methods for Euclidean space (such as images rotation, cropping, changing color, Gaussian blur, etc.) are no longer applicable. Therefore, we propose several DA methods on the SPD manifolds for the sake of synthesizing new data points to increase the sample diversity while preserving the underlying Riemannian geometry of the input data manifold. Given a SPD matrix $X \in \mathbb{R}^{d \times d}$, we can have the following:

- **Random Perturbation (RP)**: This method enhances data by directly imposing random perturbation on the SPD matrices. The specific operation can be expressed as:

$$X' = X + \Delta X, \tag{3}$$

  where $\Delta X$ represents the perturbation matrix of positive definiteness, $X'$ is the transformed matrix. We randomly generate a SPD matrix $\Delta X$ with each entry has a sufficient small value, and add it to the original matrix $X$ to realize random perturbation.

- **Tangent Perturbation (TP)**: This method exploits the diffeomorphism between the tangent space at the identity matrix and the space of SPD matrices. Specifically, the data is first mapped to the tangent space using the logarithm map, then a small perturbation is added to the tangent space. Finally, the transformed data is mapped back to the SPD manifold via the exponential map. The specific operation of TP can be formulated as:

$$X' = \text{Exp}_I(\text{Log}_I(X) + \Delta S), \tag{4}$$

  where $X'$ is the transformed matrix, $I$ is the identity matrix, and $\Delta S$ is the added symmetric perturbation matrix.

- **Matrix Scaling (MS)**: This method perturbs each SPD matrix by scaling its elements, given by:

$$X' = kX, \tag{5}$$

  where $X'$ means the scaled SPD matrix, and $k$ is a constant greater than zero.

- **Pre-SPD Perturbation Enhancement (Pre-SPE)**: Pre-SPD perturbation involves adding perturbations to the feature matrix before modeling it onto the SPD manifold. The specific operation can be expressed as:

$$X' = f_{MTS}(M + \Delta R), \tag{6}$$

where $M$ is the feature matrix generated from the original data sequence (image set, video clip, *etc*), $\Delta R$ is a random perturbation matrix, and $f_{MTS}(\cdot)$ is a function that models the input matrix onto the SPD manifold, which is actually the covariance representation.

To show the effects of the proposed RDA strategies, we take the MDSD dataset as an example to explore the impact of the various DA on the classification performance of the proposed model. Table 1 provides the accuracy of the linear evaluation under different RDA strategies. It can be found that prominent augmentation strategy is Tangent Perturbation. We speculate that, adding perturbations in the tangent space can better simulate the actual data variations of the manifold. This type of disturbance intro-

Table 1: RDA comparison on the MDSD dataset.

| RDA Methods | Accuracy (%) |
|---|---|
| RP | 28.21 |
| **TP** | **37.17** |
| MS | 35.89 |
| Pre-SPE | 30.77 |

duces new information while effectively preserving the data's core structure and characteristics, which helps enhance the model's generalization ability. Meanwhile, We speculate that random perturbation method will lead to suboptimal results, because this method ignores the non-Euclidean property of the data. While this method can preserve symmetric positive definiteness, a smaller perturbation directly into the manifold space will disrupt the data structure. Conversely, the way of adding a smaller perturbation in the tangent space does not destroy the structural information of the input data. The data may deviate from the original geodesic to some extent, but the main geometric structure infomation is preserved. In all subsequent experiments, we chose the TP strategy.

### 3.3 THE BASIC LAYERS OF SPD NEURAL NETWORK

**BiMap Layer:** This layer can be thought of as a variation of the standard dense layer, wherein the input SPD matrices are compressed into lower-dimensional ones using a bilinear mapping function $f_b$, expressed as: $X_k = f_b^{(k)}(W_k, X_{k-1}) = W_k^\top X_{k-1} W_k$, where $W_k \in \mathbb{R}^{d_k \times d_{k-1}}(d_k < d_{k-1})$ is the transformation matrix to be learned. To ensure that $X_k$ lies in another SPD manifold $S_{++}^{d_k}$, $W_k$ needs to be column full-rank. In addition, it is necessary to impose semi-orthogonality constraint on $W_k$, which results in a compact Stiefel manifold $St(d_k, d_{k-1})$ for the weight space Arsigny et al. (2007). By optimizing $W_k$ over this manifold, it becomes possible to yield optimal solutions.

**ReEig Layer:** This layer is likened to the ReLU layer in traditional ConvNets, with the aim of introducing non-linearity into SPDNet to enhance its discriminatory power, while also serving the role of eigenvalue regularization. Specifically, this layer involves using a nonlinear rectification function $f_r$ to adjust the small positive eigenvalues of each input SPD matrix, given below: $X_k = f_r^{(k)}(X_{k-1}) = U_{k-1}\max(\epsilon I, \Sigma_{k-1})U_{k-1}^\top$, where $\epsilon$ is a small activation threshold, and $X_{k-1} = U_{k-1}\Sigma_{k-1}U_{k-1}^\top$ refers to the eigenvalue decomposition. The ReEig operation can protect the matrices from degeneration, as is evident.

**LogEig Layer:** This layer is designed to enable the Euclidean learning methods to be applicable to the generated manifold-valued features. It is implemented by imposing Riemannian computation on the input SPD matrices using the logarithmic mapping function $f_l$, formulated as: $X_k = f_l^{(k)}(X_{k-1}) = U_{k-1}\log(\Sigma_{k-1})U_{k-1}^\top$. Here, $X_{k-1} = U_{k-1}\Sigma_{k-1}U_{k-1}^\top$ refers to the eigenvalue decomposition, and $\log(\Sigma_{k-1})$ denotes the logarithm operation applied to each diagonal element of $\Sigma_{k-1}$. Through this operation, traditional fully connected (FC) layers and cross-entropy loss can be introduced into the obtained flat space for visual classification.

## 3.4 CONTRASTIVE LEARNING AND INFONCE LOSS

The core idea of the InfoNCE loss function He et al. (2020) is to learn effective data representations by pulling positive sample pairs (similar or related pairs) closer and pushing negative sample pairs (dissimilar or unrelated pairs) farther apart.

- Throughout the training process, an anchor sample, a positive sample (similar or related to the anchor sample), and multiple negative samples (dissimilar or unrelated to the anchor sample) are selected.
- The encoder (SPDNet) is utilized to embed the anchor sample, positive sample, and negatives into a discriminative metric subspace anchor sample, the positive sample, and the negative samples.
- Dot product or cosine similarity is used to compute the similarity of anchor-positive sample pair and anchor-negative sample pairs, respectively.
- The goal of the InfoNCE loss is to maximize the similarity of the anchor-positive sample pair while minimizing the similarity of the anchor-negative sample pairs.

More specifically, InfoNCE Loss can be defined as:

$$\mathcal{L}_{\text{InfoNCE}} = -\log \frac{\exp\left(\text{sim}\left(x, x^+\right)/\tau\right)}{\sum_{k=1}^{K} \exp\left(\text{sim}\left(x, x_k\right)/\tau\right)}, \tag{7}$$

where $x$ represents the anchor sample, and $x^+$ represents the positive sample. We use $x^k$ ($k = 1 \to K$) to denote in the set composed by all $x^+$ and $x^-$, the logarithm of positive samples is 1 and the logarithm of negative samples is N. The sim(x, y) represents the cosine similarity between samples $x$ and $y$, $\tau$ is the Temperature Parameter, used to control the scaling of the similarity score and the sensitivity of the loss function.

## 3.5 RIEMANNIAN METRIC LEARNING

In SPD manifold neural networks, most architectures utilize only a single cross-entropy loss to supervise the entire network, overlooking the specific data distribution characteristics within and between classes during the process of learning the SPD matrix. As a consequence, the variability information conveyed by the inputs can not be encoded explicitly during training, making the resulting lower-dimensional geometric representations may not be powerful enough for improving classification. Inspired by the merits of metric learning Wang et al. (2022b), we propose a Riemannian metric learning (RML) module to improve the discriminability of the suggested model by explicitly encoding and learning the intrinsic data distributions of the input geometric features, formulated as:

$$\mathcal{L} = \frac{1}{N_{\mathfrak{A}}} \sum_{i,j \in \mathfrak{A}} \max\left(\mathcal{D}_{\text{lem}}^{\mathcal{W}_k}\left(\boldsymbol{X}_k^{i,0}, \boldsymbol{X}_k^{j,+}\right), \xi_{\mathfrak{A}}\right) \tag{8}$$

$$- \frac{1}{N_{\mathfrak{B}}} \sum_{i,j \in \mathfrak{B}} \max\left(\xi_{\mathfrak{B}} - \mathcal{D}_{\text{lem}}^{\mathcal{W}_k}\left(\boldsymbol{X}_k^{i,o}, \boldsymbol{X}_k^{j,-}\right), 0\right),$$

where $(\boldsymbol{X}_k^{i,0}, \boldsymbol{X}_k^{j,+}, \boldsymbol{X}_k^{j,-})$ is a triplet. The method of constructing triplets in this paper involves randomly selecting a sample as $\boldsymbol{X}_k^{i,o}$, within a batch, and obtaining a positive sample through RDA represented by $\boldsymbol{X}_k^{j,+}$. The remaining samples in the batch are designated as the negative samples, signified as $\boldsymbol{X}_k^{j,-}$. From Eq. (12), it can be expected that the Riemannian distance of the positive sample pair $(\boldsymbol{X}_k^{i,o}, \boldsymbol{X}_k^{j,+})$ is smaller than a manifold margin $\xi_{\mathcal{A}}$, while the Riemannian distance of the negative sample pair $(\boldsymbol{X}_k^{i,o}, \boldsymbol{X}_k^{j,-})$ is larger than $\xi_{\mathcal{B}}$. The specific form of $\mathcal{D}_{\text{lem}}^{\mathcal{W}_k}\left(\boldsymbol{X}_k^{i,0}, \boldsymbol{\Omega}\right)$ is given below:

$$\mathcal{D}_{\text{lem}}^{\mathcal{W}_k}(\mathbf{X}_k^{i,0}, \Omega) = \left\|\mathbf{W}_k^\top \log(\mathbf{X}_k^{i,o})\mathbf{W}_k - \mathbf{W}_k^\top \log(\Omega)\mathbf{W}_k\right\|_F^2, \tag{9}$$

Table 2: Accuracy(%) of different algorithms on the Virus, MDSD, YTC and AFEW datasets.

| Method | Virus | MDSD | YTC | AFEW |
|---|---|---|---|---|
| SPDML-AIM Harandi et al. (2017) | 40.68 | 30.04 | 64.66 | 26.72 |
| SPDML-Stein Harandi et al. (2017) | 42.00 | 27.69 | 61.57 | 24.55 |
| GrNet Huang et al. (2018) | 39.33 | 31.32 | 70.68 | 34.23 |
| SPDNet Huang & Van Gool (2017) | 53.33 | 32.05 | 67.38 | 34.23 |
| SPDNetBN Brooks et al. (2019) | 36.67 | 35.89 | 67.37 | **36.20** |
| MRMML Wang et al. (2022a) | 56.67 | 36.67 | 73.82 | 35.71 |
| GEPML Wang et al. (2022d) | N/A | 35.33 | 73.45 | 33.78 |
| SymNet Wang et al. (2022c) | **71.53** | 35.58 | 71.46 | 31.89 |
| GDLNet Wang et al. (2024) | 60.00 | 36.67 | 71.63 | 33.42 |
| **SymCL (Ours)** | 56.67 | **37.17** | **74.82** | 33.42 |

where $\Omega$ denotes $\boldsymbol{X}_k^{j,+}$ or $\boldsymbol{X}_k^{j,-}$. In this paper, the proposed RML module is trained in conjunction with the previously mentioned InfoNCE loss.

## 3.6 PRETRAINING

In this paper, we first use the proposed contrastive learning method to pre-train a SPD encoder. Then, a fully connected (FC) layer is appended to the tail of the trained SPD encoder for downstream tasks. We use the proxy mechanism described in the previous proxy section to construct positive and negative sample pairs, and pre-train the entire network with both InfoNCE Loss and RML (Riemannian triplet loss). We know that in the realm of traditional, contrastive learning the encoders are typically pre-trained in a self-supervised manner using the ImageNet dataset, followed by supervised fine-tuning on the corresponding dataset for downstream tasks. However, in the context of Riemannian networks, we lack a universal large-scale dataset similar to ImageNet, making it impractical to conduct general pre-training. Moreover, the eigenvalue operation involved in SPDNet primarily relies on the CPU and cannot be accelerated by GPU, making the computation very time-consuming. Even though it is feasible to model ImageNet on the SPD manifold for training, the large-scale dataset necessitates a deeper network. Riemannian neural network models suffer from the degradation of structural information during the multi-level transmission of data. Simply stacking more layers to increase the network depth in shallow Riemannian networks does not alleviate this problem. This paper only discusses the feasibility and effectiveness of the proposed Riemannian CL on the SPD manifolds. To the best of our knowledge, this is the first work on the extension of CL to the scenario of Riemannian manifolds.

## 4 EXPERIMENT

In this section, we evaluate the effectiveness of the proposed method on different visual classification tasks, namely the video-based emotion recognition using the AFEW dataset, the dynamic scene classification using the MDSD dataset, the YTC dataset and the cell identification task using the Virus dataset.

### 4.1 DATASET DETAILS

In all the following experiments, we set up as follows: 1) when modeling the original data onto the SPD manifold, the regularization parameter $\varepsilon$ is set to 1e-5. 2) Our designed network model consists of three building blocks, each block comprising a BiMap layer and a ReEig layer. We set the dimensions of the weight matrices in the model to $400 \times 200$, $200 \times 100$ and $100 \times 50$. 3) For both of pre-training and downstream fine-tuning, we use the Adam optimizer for parameter updates. Please refer to the appendix A.2 for a detailed description of each dataset and specific settings.

## 4.2 RESULTS

Referring to the classification results in Table 2, some interesting experimental findings can be drawn. Firstly, SymCL achieves higher recognition accuracy on both YTC and MDSD datasets compared to SPDNet trained under the supervised learning paradigm, indicating that applying the CL paradigm to the training process of manifold networks helps enhance the discriminative capability of the learned features. Moreover, the classification performance of the AFEW dataset is not as good as SPDNet. The main reason lies in the fact that AFEW has significant intra-class variability and inter-class similarity. To achieve high recognition accuracy, the model needs to capture key facial expression changes without being distracted by background and appearance information. However, in this study, positive and negative samples are constructed using Riemannian data augmentation, which may result in the similarity between positive samples being primarily reflected in appearance and background information rather than expressions, thereby affecting the model's discriminative learning ability. According to the experimental results reported in Table 2, we can observe that on the Virus dataset, SymNet outperforms our method. This is mainly because the Virus is a small scale dataset, which has been verified to be more suitable for the lightweight SymNet. Additionally, each image on the Virus dataset has an irregular appearance, which causes subspace-based methods to lose some useful information during representation learning. On the AFEW dataset, SPDNetBN achieved the highest accuracy. We know that in standard deep neural networks, batch normalization improves training stability and accelerates convergence by normalizing activations. This concept is applied to SPDNet in the form of Riemannian Batch Normalization (RBN), resulting in SPDNetBN. This method not only normalizes data in a way that respects the manifold's geometric, but also centralizes and biases SPD matrices using parallel transport and the Riemannian barycenter. This allows SPDNetBN to demonstrate superior classification performance.

Secondly, it can be intuitively observed that SymCL surpasses most of the comparative algorithms (all of them are supervised) on the four used datasets. This is invaluable for self-supervised learning. This validates that the proposed self-supervised contrastive learning mechanism on the Riemannian manifold enables the SPD encoder to capture and learn more authentic data structures. Moreover, the introduced RML module, by explicitly encoding the intra-class and inter-class distribution of the data, helps in training a more discriminative Riemannian network embedding. In this context, a more suitable classification hypersphere can be obtained, which enhances classification accuracy.

## 4.3 ABLATION EXPERIMENTS

### 4.3.1 ABLATION STUDY ON THE ROBUSTNESS OF THE SYMCL

The various experimental results mentioned above indicate that the SymCL proposed in this paper has certain advantages in improving the accuracy of image set classification compared to some representative methods. To further assess the robustness of the models trained using CL, this section conducts experiments on the MDSD dataset. To investigate the impact of erroneous labels on the model accuracy, Specifically, we first randomly mislabel the data according to a certain proportion, then separately

Table 3: Accuracy(%) of SPDNet and SymCL under different error rate on the MDSD dataset.

| Error (%) | SPDNet | SymCL |
|-----------|--------|-------|
| 3 | 23.08 | 28.21 |
| 5 | 20.51 | 25.64 |
| 10 | 15.38 | 20.51 |

evaluate the classification performance of the supervised learning SPDNet (SPDNet-SL) and SymCL under these conditions. From Table 3, several interesting experimental observations can be found. Firstly, after mislabeling 3%, 5%, and 10% of the labels, the accuracy of SPDNet dropped from 32.05% (before mislabeling) to 23.08%, 20.51%, and 15.38%, respectively. This demonstrates that supervised learning methods are very sensitive to errors in the labels. According to the SymCL classification results listed in Table 3, we can observe that, at the same error rate, the classification accuracy of SymCL is significantly higher than that of SPDNet on the MDSD dataset. Upon analysis, we speculate that in the SymCL algorithm, the mislabeled labels primarily affect the classifier training phase during linear evaluation. However, the erroneous labels do not impact the self-supervised pre-training phase, the model has already learned effective data representations in this phase. There-

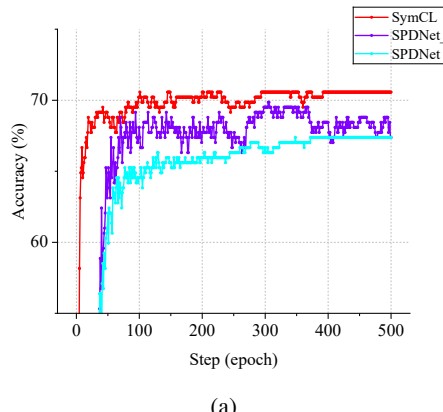 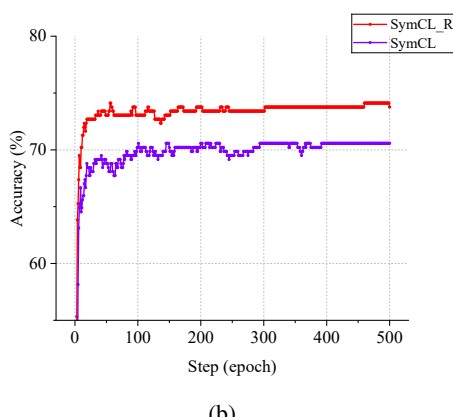

(a)                                                        (b)

Figure 2: (a) Accuracy comparison of three methods: supervised learning SPDNet (SPDNet), supervised learning SPDNet with RML (SPDNet_R) and contrastive learning SPDNet (SymCL). (b) Accuracy comparison of two methods: SymCL and SymCL with RML (SymCL_R).

Table 4: Accuracy comparison (%) on the Virus, MDSD, YTC, and AFEW datasets.

| Methods | Virus | MDSD | YTC | AFEW |
|---------|-------|------|-----|------|
| SPDNet | 50.00 | 32.05 | 65.96 | 34.23 |
| **SPDNet-RML** | **53.33** | **33.33** | **69.86** | **35.31** |
| SymCL | 55.33 | 35.63 | 72.34 | 32.88 |
| **SymCL-RML** | **56.67** | **37.17** | **74.82** | **33.42** |

fore, under identical circumstances, the erroneous labels have relatively less impact on the accuracy of SymCL. The above experimental results demonstrate the robustness of self-supervised CL.

### 4.3.2 ABLATION STUDY OF RIEMANNIAN METRIC LEARNING MODULE

In this part, we conduct experiments on the AFEW, MDSD, Virus, and YTC datasets to study the impact of the RML module on the classification performance of the proposed method. The experimental results under different conditions are shown in Table 4. In this table, SPDNet-RML represents the integration of the RML module into the tail of the last BiMap layer in the network. Placing the RML after the previous two set of SPD blocks (BiMap + ReEig) can provide a more effective structured representation of the original data with richer semantic information, thereby better training the entire network together with the cross-entropy loss. The first and second rows in Table 4 list the classification performance of the original SPDNet and SPDNet with the RML module under a supervised paradigm. We choose the YTC dataset to plot model accuracy. Figure 2a contrasts the accuracy of SPDNet, SPDNet-RML and SymCL, Figure 2b contrasts the performance of SymCL and SymCL with RML module (SymCL-RML). The third and fourth rows in Table 4 compare the performance of the pre-trained models with and without using the RML under a self-supervised CL paradigm. It can be noted that the inclusion of RML can enhance the classification performance of the network regardless of whether it is trained using the supervised or self-supervised paradigm. Furthermore, we can find that regardless of the presence of the RML module, models pre-trained using CL show higher accuracy in downstream tasks on all the used datasets, except the AFEW dataset. We speculate that the original supervised learning-based method (SPDNet) tends to be easily affected by label noise. On the contrary, our proposed SymCL is based on the intrinsic geometric distribution of the data, and learns more discriminative information with the help of metric learning mechanism, thereby improving the model performance, qualifying it to produce new feature manifolds with improved discriminability. Overall speaking, the InfoNCE loss explicitly utilizes the self-supervised signal information and implicitly utilizes the data distribution; in contrast, RML explicitly leverages the data distribution distribution while implicitly leverages the label information, forming a dynamic complementary relationship with the InfoNCE loss.

## 5 CONCLUSION

In this paper, we extend the traditional contrastive learning paradigm to the Riemannian manifolds, paving the way for self-supervised learning in non-Euclidean spaces. For the proposed method, we design a novel proxy mechanism for constructing positive and negative samples, and propose four different DA methods tailored for Riemannian manifolds. Furthermore, we introduce a RML module and integrate it with InfoNCE loss to facilitate training an improved network embedding. The effectiveness of the proposed method is investigated through evaluations on four benchmarking datasets, with additional ablation studies underscoring the contribution of each individual component.

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

# A  APPENDIX

## A.1  DERIVATION OF LEM FORMULA

A family of $d$-by-$d$ SPD matrices is a commutative Lie group, with a manifold structure denoted as $S^d_{++}$. More formally:

$$S^d_{++} := \{X \in \mathbb{R}^{d \times d} : X = X^T, \nu^T X \nu > 0, \forall \nu \in \mathbb{R}^d \setminus \{0_d\}\}. \tag{10}$$

Therefore, the concepts of differential geometry, such as geodesic, can be utilized to address $S^d_{++}$. Moreover, any bi-invariant metric $\langle,\rangle$ on the Lie group of SPD matrices corresponds to an Euclidean metric in the SPD matrix logarithmic domain, i.e., the tangent space at identity matrix $T_I S^d_{++}$, it is also called the Log-Euclidean Metric.

Specifically, for any two tangent elements $T_i, T_j$, their scalar product in $T_X S^d_{++}$ is given by:

$$\langle T_i, T_j \rangle_X = \langle D_X \log .T_i, D_X \log .T_j \rangle, \tag{11}$$

where $D_X \log .T$ signifies the directional derivative of the matrix logarithm at $X$ along $T$. The logarithmic mapping with respect to the Riemannian metric is defined by the matrix logarithms:

$$\log_{X_i}(X_j) = D \log(X_i) \exp .(\log(X_j) - \log(X_i)). \tag{12}$$

On the basis of the differentiation of $\log \circ \exp = I$, we can obtain $D_{\log_{\exp}} = (D_X \log .)^{-1}$. Similarly, the matrix exponential mapping can be expressed as:

$$\exp_{X_i}(T_j) = \exp(\log(X_i) + D_{X_i} \log .T_j). \tag{13}$$

Combining Eq. (2), Eq. (3), and Eq. (4), the LEM can be formulated as:

$$D_{LEM}(X_i, X_j) = \| \log(X_j) - \log(X_i) \|_F. \tag{14}$$

It can be found that compared to the Affine-Invariant Riemannian Metric (AIRM), the Log-Euclidean Metric (LEM) works directly in the logarithmic domain of SPD matrices, offering higher computational efficiency.

## A.2  DATASETS DESCRIPTION AND SETTING

### A.2.1  YTC DATASET

This dataset was collected from the YouTube website, containing a total of 1,910 video clips, belonging to 47 different categories. Each video clip consists of hundreds of facial images, and there exists a considerable variability within the same category in terms of expression, illumination, obstructions, resolution, and posture, making classification on this dataset quite challenging. For pre-training, we set the batch size and learning rate to 64 and 6e-4 respectively on the YTC dataset. For downstream fine-tuning, we set the learning rate to 1e-3.

### A.2.2  VIRUS DATASET

This dataset was collected using Transmission Electron Microscopy (TEM) technology and contains 1500 TEM images belonging to 15 different virus categories. This dataset exhibits a wide range of intra- and inter-class variations, primarily lying in two aspects: 1) The shapes of the viruses vary from polygons to icosahedrons; 2) The diameters of the viruses range from 25 nanometers to 270 nanometers. These make cell identification on this dataset quite difficult. For pre-training, we set the batch size and learning rate to 32 and 3e-4 respectively on the YTC dataset. For downstream fine-tuning, we set the learning rate to 1e-3.

Table 5: Accuracy comparison (%) on the Virus, MDSD, YTC, and AFEW datasets.

| Methods | Virus | MDSD | YTC | AFEW |
|---|---|---|---|---|
| SPDNet | 50.00 | 32.05 | 65.96 | 34.23 |
| SymCL (TP) | **55.33** | **35.63** | **72.34** | 32.88 |
| **SymCL (DT)** | 53.33 | 35.63 | 71.28 | **33.42** |

### A.2.3 MDSD Dataset

This dataset includes 13 different dynamic scene categories, each category makes up of 10 video clips. As this dataset is collected in real-world scenario with significant variability in appearance, resolution, and physical form. it is quite challenging for scene classification. For pre-training, we set the batch size and learning rate to 32 and 3e-4 respectively on the YTC dataset. For downstream fine-tuning, we set the learning rate to 1e-3.

### A.2.4 AFEW Dataset

This dataset contains 1,345 video clips belonging to 7 types of facial expressions. Since these video sequences are collected from movies, featuring content scenes close to the real-world scenario, A notable characteristic of this dataset is that it poses a large intra-class diversity and inter-class ambiguity. For pre-training, we set the batch size and learning rate to 32 and 3e-4 respectively on the YTC dataset. For downstream fine-tuning, we set the learning rate to 1e-3.

### A.3 New Data Augmentation Methods

In this study, we propose a novel data augmentation strategy aimed at enhancing the generalization ability of encoder models (for reader's reference only). The specific approach is as follows: a dropout layer is introduced at the final stage of the encoder to increase randomness during the model training process. In each batch, one sample is randomly selected for two forward passes, generating two different tensors. These two tensors are treated as a positive sample pair for training the model to recognize similar features. Meanwhile, the other samples in the batch are considered negative sample pairs, helping to strengthen the model's ability to distinguish between different features (This strategy is hereinafter referred to as DT: dropout twice.). This strategy effectively improves the model's performance and robustness in complex environments by introducing positive and negative sample contrasts within the same sample. Since the proposed data augmen-

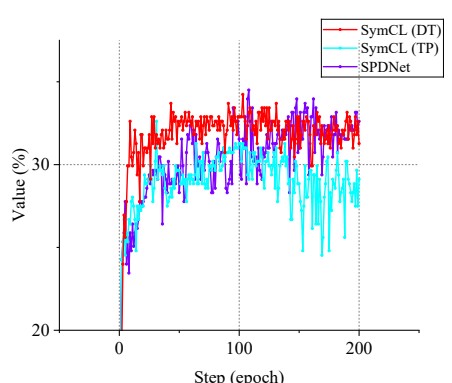

Figure 3: The red line indicates SymCL with DT data augmentation strategy, the cyan line indicates SymCL with TP data augmentation strategy, the purple line indicates SPDNet.

tation strategy was not implemented at the Riemannian manifold level, this paper will not discuss it in detail in the main text. The corresponding technical details and implementation will be described thoroughly in the appendix. Additionally, because the construction of positive and negative samples is positioned at the end of the encoder, the RML module located on the last BiMap layer will no longer be activated. Therefore, we will only compare the effects of the new data augmentation strategy with the Tangent Perturbation (TP) strategy discussed in the main text under the same experimental conditions in SymCL (without RML module). The experimental results under different data augmentation strategies are shown in Table 5. We selected the AFEW dataset to plot the convergence curve; please refer to Figure 3. We can observe that on some datasets (such as AFEW), the linear evaluation accuracy of the DT data augmentation strategy is higher than that of TP. The main reason is that, positive and negative samples are constructed using Riemannian (TP) data aug-

mentation, which may result in the similarity between positive samples being primarily reflected in appearance and background information rather than expressions, thereby affecting the model's discriminative learning ability. In summary, although the DT strategy may be more suitable for the AFEW dataset, the overall performance indicates that the DT strategy does not achieve the optimal solution across all classification tasks.

