# OpenReview forum: "SymCL: Riemannian Contrastive Learning on the Symmetric Positive Definite Manifold for Visual Classification"
_ICLR.cc/2025/Conference — ICLR 2025 Conference Withdrawn Submission_

### Official Review · Reviewer_wiE7 · 2024-10-30

**Soundness:** 2
**Presentation:** 2
**Contribution:** 2
**Rating:** 3
**Confidence:** 4

**Summary:**

ymmetric positive definite (SPD) is an effective technique for data information extraction, which encodes spatiotemporal statistical information with a curved Riemannian manifold. To utilize the property, the paper combines SPD and contrastive learning (CL) together to form a framework. A Riemannian triplet loss-base loss is used for the proposed method. Some experiments have been conducted to verify the performance of the proposed method.

**Strengths:**

1. A combination is given by using SPD and CL.
2. A novel Riemannian triplet loss is introduced for the proposed method.
3. Some data augmentation methods are proposed.

**Weaknesses:**

1.	The proposed method is a combination of SPD and CL, using a different metric for the loss function. The novelty is not enough. Why did the authors introduce SPD into contrastive learning without providing a good motivation, but now it seems that it's just a combination of the two.
2.	Figure 1 directly shows the learning process of the proposed method, from which we can see that the proposed method just is RML adds CL.
3.	Some definitions of letter variables are not provided, and the algorithm flow for this method is also not given, such as what are the inputs and outputs? The designed four augmentation methods cannot be considered as a major contribution.
4.	Many experimental details were not provided, such as experimental settings, experimental environment configuration, and comparison methods.
5.	The parameter sensitivity analysis and convergence analysis have not been provided.
6.	Many experimental effect analyses have not been provided, such as visualization effects.
7.	In the experiments, no compared methods are used, how to effectively validate the SOTA performance of this method?

**Questions:**

Please see the weaknesses.

---

### Official Review · Reviewer_2K7G · 2024-10-30

**Soundness:** 2
**Presentation:** 3
**Contribution:** 1
**Rating:** 3
**Confidence:** 2

**Summary:**

This paper transfers contrastive learning from Euclidean space to Riemannian manifolds. They replace Euclidean similarities in the contrastive loss with Riemannian distances. They use four datasets to verify the effectiveness.

**Strengths:**

This paper writes a clear background.

**Weaknesses:**

1. The motivation and advantages of this method are not clear. why do we need the contrastive learning loss for SPD manifolds? The label noise is not a special issue for SPD manifolds.

2. The novelty is limited. It seems that this paper simply replaces the Euclidean similarity with Riemannian distances.

3. The experiments are too weak. Only using four datasets that are somewhat out of data.

4. What is the application of this method? How can the SPD contrastive learning help the community?

**Questions:**

See the above weakness.

---

### Official Review · Reviewer_Rytf · 2024-11-02

**Soundness:** 2
**Presentation:** 2
**Contribution:** 2
**Rating:** 5
**Confidence:** 5

**Summary:**

This paper argues that the inherent dependence on labels in supervised learning makes learning on SPD manifold vulnerable to the effect of label noise. Therefore, the authors apply contrastive learning on SPD manifold. Meanwhile, they believe that traditional data augmentation methods are not suitable for SPD manifold, and introduce four data augmentation techniques. Additionally, a RML loss is combined to train SPD networks. The experiments are conducted on four different datasets.

**Strengths:**

+: The paper tries to apply contrastive learning on SPD manifold, and introduces some data augmentation techniques.

+: Experiments show the proposed method achieves good performance.

**Weaknesses:**

-: The technical novelty of the proposed method seems limited.

Firstly, the original contrastive learning is directly applied on SPD manifold without clear changes or improvement. Besides, this paper claims that traditional augmentation methods are unsuitable for SPD networks. However, no baseline is compared.

Secondly, the authors should clarify that what are differences between the proposed RML loss and work [1]? Besides, RML loss shares a similar concept with InfoNCE, where both of them aim to bring anchor samples closer to positive samples and push them further from negative samples. Therefore, why they are complementary?

-: The experimental descriptions seem not very clear.

Specifically, it is unclear which dataset was used for pre-training, the detailed setup of the pre-training phase, and the approach taken for training on downstream tasks. For example, is the entire network trained, or is only the classifier fine-tuned?

Moreover, based on the experimental results, the performance improvements brought by the proposed method are not significant. The performance of the MDSD and AFEW datasets is inferior to the results mentioned in the paper [1], yet these findings are not clearly discussed in the current paper.

The paper explains that the proposed method performs poorly on the Virus dataset as "This is mainly because the Virus is a small-scale dataset, which has been verified to be more suitable for the lightweight SymNet."  Therefore, does basic backbone heavily affect the final performance? If so, the authors would better compare different methods by using the same basic backbone.

[1] WANG R, WU X J, CHEN Z, et al. Learning a discriminative SPD manifold neural network for image set classification[J/OL]. Neural Networks, 2022: 94-110.

**Questions:**

1.	It is unclear whether the compared methods utilize the same basic backbone in Tab 2? What are the parameter numbers of different methods?

2.	The author would better compare the proposed data augmentation methods with traditional one in the image space.

3.	The author would better analyze and compare the differences between contrastive learning loss and RML loss.

---

### Official Review · Reviewer_h57s · 2024-11-03

**Soundness:** 3
**Presentation:** 3
**Contribution:** 1
**Rating:** 5
**Confidence:** 5

**Summary:**

This submission proposed a method combining Symmetric Positive Definite (SPD) as feature descriptor into contrastive learning (CL) for classification. A Riemannian triplet loss is proposed to guide the training process. Experiments are conducted on four datasets including AFEW, MDSD, YTC, and the Virus.

**Strengths:**

This submission is well written and easy to follow. The idea is clear. Experiments are conducted on multiple tasks.

**Weaknesses:**

The idea is very straightforward by combining existed SPD and CL. Novelty is very limited. Although some ablation studies are performed, it still need to have some comparison to SOTA methods.

**Questions:**

As mentioned weakness part, the submission need to have some comparison studies to show its advantage.

---

### Note · Authors · 2024-12-30

I have read and agree with the venue's withdrawal policy on behalf of myself and my co-authors.